# Characterization of the Effects of a Novel Probiotic on *Salmonella* Colonization of a Piglet-Derived Intestinal Microbiota Using Improved Bioreactor

**DOI:** 10.3390/ani14050787

**Published:** 2024-03-02

**Authors:** Amely Grandmont, Mohamed Rhouma, Marie-Pierre Létourneau-Montminy, William Thériault, Isabelle Mainville, Yves Arcand, Roland Leduc, Bruno Demers, Alexandre Thibodeau

**Affiliations:** 1Chaire de Recherche en Salubrité des Viandes, Département de Microbiologie et Pathologie, Faculté de Médecine Vétérinaire, Université de Montréal, Montreal, QC J2S 2M2, Canada; amely.grandmont@umontreal.ca (A.G.);; 2Groupe de Recherche et d’Enseignement en Salubrité Alimentaire, Faculté de Médecine Vétérinaire, Université de Montréal, Montreal, QC J2S 2M2, Canada; 3Centre de Recherche en Infectiologie Porcine et Avicole, Faculté de Médecine Vétérinaire, Université de Montréal, Montreal, QC J2S 2M2, Canada; marie-pierre.letourneau-montminy.1@ulaval.ca; 4Chaire de Recherche sur les Stratégies Alternatives d’Alimentation des Porcs et des Volailles: Approche Systémique pour un Développement Durable, Faculté des Sciences de l’Agriculture et de l’Alimentation, Université Laval, Quebec, QC G1V 0A6, Canada; 5Agriculture et Agroalimentaire Canada, St-Hyacinthe, QC J2S 8E3, Canada; isabelle.mainville@agr.gc.ca (I.M.); yves.arcand@agr.gc.ca (Y.A.); 6NUVAC Éco-Sciences, Valcourt, QC J0E 2L0, Canada; roland.leduc@nuvac.ca (R.L.); bruno.demers@nuvac.ca (B.D.)

**Keywords:** probiotic, *Bacillus*, *Salmonella*, pig, microbiota, bioreactor

## Abstract

**Simple Summary:**

*Salmonella* is a pathogen of worldwide public health concern, responsible for gastroenteritis and sometimes other health complications. Humans can be infected by eating contaminated pig products. On farms, pigs are usually asymptomatic carriers of *Salmonella*, making it difficult to detect the disease clinically. Some methods for controlling this microorganism are used on the farm, but their effectiveness varies between studies. Probiotics could be a reliable option in this context, reducing the presence of *Salmonella* in the pig’s intestinal tract while modulating its microbiota.

**Abstract:**

The carriage of *Salmonella* in pigs is a major concern for the agri-food industry and for global healthcare systems. Humans could develop salmonellosis when consuming contaminated pig products. On the other hand, some *Salmonella* serotypes could cause disease in swine, leading to economic losses on farms. The purpose of the present study was to characterize the anti-*Salmonella* activity of a novel *Bacillus*-based probiotic using a bioreactor containing a piglet-derived intestinal microbiota. Two methods of probiotic administration were tested: a single daily and a continuous dose. *Salmonella* enumeration was performed using selective agar at T24h, T48h, T72h, T96h and T120h. The DNA was extracted from bioreactor samples to perform microbiome profiling by targeted 16S rRNA gene sequencing on Illumina Miseq. The quantification of short-chain fatty acids (SCFAs) was also assessed at T120h. The probiotic decreased *Salmonella* counts at T96 for the daily dose and at T120 for the continuous one. Both probiotic doses affected the alpha and beta diversity of the piglet-derived microbiota (*p* < 0.05). A decrease in acetate concentration and an increase in propionate proportion were observed in the continuous condition. In conclusion, the tested *Bacillus*-based product showed a potential to modulate microbiota and reduce *Salmonella* colonization in a piglet-derived intestinal microbiota and could therefore be used in vivo.

## 1. Introduction

*Salmonella* is a major cause of foodborne diseases of bacterial etiology in humans worldwide, with 88,000 cases approximately in Canada and about 1.35 million cases, 26,500 hospitalizations, and 420 deaths in the United States every year [1,2,3]. In humans, salmonellosis is associated with symptoms such as diarrhea, nausea, stomach cramp, and fever. It can eventually lead to death if patients develop bacteremia [4]. Food-producing animals, particularly poultry, pigs, and cattle, are the main reservoirs of the different *Salmonella* serotypes associated with human salmonellosis [5]. Human contamination from pig products is estimated to be between 15 and 23% of all human salmonellosis cases in Europe [6]. It is noteworthy that pigs are generally asymptomatic carriers of *Salmonella* making it difficult to detect *Salmonella* in these animals [7]. Some *Salmonella* serovars such as *S. enterica* serovar Cholerasuis and *S. enterica* serovar Typhimurium can cause disease in pigs with a clinical presentation characterized by diarrhea, dehydration, fever and sometimes death [8,9]. The selection of antimicrobial resistance (AMR) among the different *Salmonella* serovars is a serious concern for human health and pig production [10]. Thereby, various actions have been carried out to limit the spread of AMR bacteria and to preserve the effectiveness of antimicrobials [11]. In this context, several alternatives (e.g., organic acids, essential oils) to the use of antimicrobials have been assessed at the farm level for the control of *Salmonella* spp. [12]. However, the results of these studies reported variability in the effectiveness of these measures, making further research on the subject essential [13,14].

A promising on-farm control measure of *Salmonella* is the use of probiotics. Indeed, probiotics are defined as “live microorganisms which, when administered in adequate amounts, confer a health benefit on the host’’ according to the World Health Organization (WHO) [15]. These products have modulating effects on the pig’s microbiota and immunity and some of them may produce bacteriocins effective against certain pathogens [16]. Probiotics have been assessed in many studies to characterize their effect on *Salmonella* and pig’s performances (e.g., average daily gain, feed conversion index) [17,18,19]. In this context, *Bacillus* spp. has been one of the most frequently explored bacteria [20]. Indeed, Larsen et al. showed an antibacterial effect of *B. amyloliquefaciens*, *B. subtilis* and *B. mojavensis* against a panel of pathogens that included *Salmonella* [21]. Some products that are already commercially available use *Bacillus*-based products (e.g., BioPlus 2B) to improve the zootechnical performance of animals and showed a diminution of approximately 4 log CFU of *Salmonella* per gram of feces while improving the animal’s feed conversion ratio [22]. In another study, the use of the same product was associated with a diminution of ammonia in swine slurry [23]. *Bacillus* spp. are also recognized as strong enzyme producers that can be used in many domains such as agri-food and agricultural sectors [24]. Indeed, they produce, for example, amylases and proteases that could increase the digestibility of the nutrients in the intestinal tract. In a study by Amhed, S. et al., a *Bacillus*-based probiotic showed an increase in digestibility for crude protein, crude fat and dry matter [22].

Inhibitory and microbiota modulation effects of probiotics have been studied both in vitro and in vivo [24]. The in vitro methods are mainly employed to describe the effect of the probiotic against one bacterial strain at a time while, for the in vivo studies, many animals are necessary to characterize the impact of probiotics on the intestinal microbiota due to the animal intestinal tract variability, especially when multiple doses or formulations of probiotics must be tested. An alternative method to study probiotics is by using a bioreactor, an in vitro laboratory-scale technology. It consists of one or many reactors that allow fermentation reactions where parameters such as pH, temperature or atmosphere environment can be controlled [25]. Such a system can cultivate and maintain a microbiota derived from the intestinal microbiota of piglets [26]. Bioreactors are an intermediate between in vitro hypothesis testing on single microorganism cultures and in vivo testing in animals.

Bioreactors have been used to study with success the impact of probiotics on a given microbiota. For instance, Cordonnier et al. determined the survival of *Saccharomyces cerevisiae*, as a probiotic, in a human semi-continuous bioreactor and analyzed its effect on the human microbiota [27]. Another study from Westfall et al. elaborated and used a complex bioreactor system to evaluate the microbiota change in the human intestinal tract using probiotics [28]. In pigs, a bioreactor system, Pigut IVM, was developed to study the colonic segment of the intestinal tract with the goal of assessing the effects of probiotics on the microbiota [29]. Another bioreactor named PolyFermS has been used to assess the effects of a *Bifidobacterium thermophilum* strain, a probiotic, on the swine proximal colon-derived microbiota as well as on *Salmonella* [30]. More recently, in our laboratory, such a system was set up [26]. Bioreactors represent non-invasive tools used to test multiple hypotheses and develop new probiotics in vitro, greatly reducing the need for animal studies. If the hypotheses can be validated in a bioreactor, they have a better chance of success when tested in animals. With the advancements in sequencing technology, it is possible to describe the microbiota diversity within the bioreactor using 16S rRNA amplicon sequencing [31]. On the other hand, it is also possible to study the microbiota function by looking at by-product productions, such as by quantifying the energy and composition of the digesta or short-chain fatty acid (SCFA) production [32]; SCFAs are a source of energy for the host and can also modulate microbiota [33]. For example, it has been shown that acetate, propionate and butyrate are inhibitors of *Salmonella* [34].

Therefore, the current study aimed to (1) evaluate the anti-*Salmonella* effect of a newly developed multi-strained *Bacillus*-based liquid product named PRO-P2702 in a bioreactor using two different methods of administration; and (2) to characterize the effect of this product on the bioreactor microbiota, the total gross energy, total crude protein, and the SCFAs production.

## 2. Materials and Methods

### 2.1. Probiotic Description

The probiotic named PRO-P2702 is a product from NUVAC Eco-Sciences (Valcourt, QC, Canada). It is a liquid probiotic containing *Bacillus* strains designed for pig consumption.

### 2.2. Disk Diffusion Assay of PRO-P2702 against Salmonella Typhimurium, Escherichia coli, Staphylococcus aureus and Enterococcus faecalis

To determine the inhibitory potential of the probiotic product against individual indicator bacteria, 10 mL of the PRO-P2702 was centrifuged at 13,000 rpm for 20 min. The supernatant and the pellet were kept and tested separately to assess if the bacteria (pellet) or the solution (supernatant containing fermentation and stabilizing products) were effective in inhibiting the selected bacteria. The supernatant was filtered with a 0.22 µm filter (SARSTEDT, Nümbrecht, Germany) prior to being used. The pellet was washed once in saline (0.85%), and then resuspended in 10 mL of saline. The supernatant and the pellet were diluted to 1/2, 1/5, 1/10 and 1/100 in PBS (OXOID Ltd., Basingstoke, UK). Mueller-Hinton agars (Becton, Dickinson and Company, Le Pont de Claix, France) were inoculated using a sterile polyester swab (Puritan Medical Products Company LLC, Guilford, ME, USA) with either a strain of *Salmonella* Typhimurium DT104 4393 resistant to rifampicin (*Salmonella* Typhimurium-R), *Escherichia coli* ATCC 25922, *Staphylococcus aureus* ATCC 25923 or *Enterococcus faecalis* ATCC 29212. For all indicator bacteria, strains were grown overnight on blood-agar (OXOID Ltd., Nepean, ON, Canada) and were suspended to 0.5 Mc Farland in PBS. Sterile disks (Benex Limited, Dublin, Ireland) were placed on the inoculated Mueller-Hinton agar. For the supernatant, 10 µL of each concentration was separately added to the paper disks. For the pellet dilutions, 10 µL of each pellet suspension was added in 0.5 cm wells made in the inoculated Mueller-Hinton agars. The inhibition zones around the tested products were observed after an incubation of 24 h at 37 °C. The diameter of the inhibition zones was measured in mm.

### 2.3. PRO-P2702 Broth Inhibition Test against Salmonella Typhimurium

To determine the bactericidal or bacteriostatic effect of the product against *Salmonella* and to characterize the inhibition dynamic of the product against different concentrations of *Salmonella*, *Salmonella* Typhimurium-R was grown from frozen stock on blood agar overnight at 37 °C. Then, 20 mL of PRO-P2702, the equivalent of 10^10^ total CFU, was put in contact with 10^3^ or 10^6^ CFU of *Salmonella* Typhimurium-R in 50 mL of brain–heart infusion (BHI) broth (Becton, Dickinson and Company, Le Pont de Claix, France) and incubated at 37 °C in a rotatory incubator (Orbital Shaker, Thermo Scientific, Waltham, MA USA, USA) set at 180 rpm. Samples were taken at T0, after 10 min, 24 h, 48 h and 120 h to assess the absence or presence of *Salmonella* Typhimurium-R by using BG Sulfa Agar (BGS) (Becton, Dickinson and Company, Le Pont de Claix, France) supplemented with rifampicin at a concentration of 0.025 mg/mL and novobiocin at a concentration of 0.02 mg/mL (BGS novo-rifampicin).

### 2.4. Recovery of Intestinal Contents for the Bioreactor’s Inoculum

#### 2.4.1. Animal Experimentation

The experimental protocol was accepted by the ethics committee of the Faculty of Veterinary Medicine of Université de Montréal with certificate number 21-Rech-2074. Eight post-weaning piglets (21 days old) were purchased from a local producer and received in the animal facility at the Faculty of Veterinary Medicine of Université de Montréal in Saint-Hyacinthe, Quebec. They were given a standard commercial feed formulation without antibiotic. Animals had access ad libitum to water, food, and environment enrichment toys. They were raised on a litter made of wood shavings. After 2 weeks of acclimation, they were euthanized to recover their intestinal contents. Caecal, ileal and colon intestinal segments for each animal were sealed off using tie wraps, then, were put in individual bags and brought to the laboratory on ice.

#### 2.4.2. Inoculum’s Recovery

At the laboratory, intestinal segments were sterilely emptied. One composite sample (all animals pooled together) from each segment was constituted. Freezing medium made as described in Bellerose, M. et al. was added in a 1:1 proportion and then the intestinal content was stomached for 1 min to obtain a homogenous texture [26]. Portions of 20 g from the colon were put in sterile tubes and then flash-frozen in liquid nitrogen. The samples were kept in a freezer at −80 °C until used. The experiment was conducted as fast as possible to minimize exposure of the intestinal content to oxygen.

### 2.5. Bioreactor Experiments

#### 2.5.1. Bioreactor Description

The bioreactor used in the current study was composed of six jacketed reactors: the first one, named the mother reactor (R1), may contain up to 600 mL and the other five reactors, named daughter reactors (R2–R6), can contain up to 300 mL each. This system was used in a previous study with the following modifications: the time used to fill the mother reactor was 24 h; during the first 24 h, a mix of anaerobic gas (90% nitrogen, 5% CO_2_ and 5% H_2_) was added to the system instead for the first 24 h and then nitrogen was used for the rest of the experiment; the gas was bubbled in the culture media 30 min prior to filling in order to keep the culture media oxygen concentration as low as possible; inulin was added to the culture media to increase nutrient for fermentative bacteria; the culture media, when made, was pooled and distributed into fractions and 20 g of inoculum was used [26].

The culture media, consisting of an in vitro digestion of piglet’s standard non-antibiotic feed, was prepared at the St-Hyacinthe Research and Development Centre (RDC) as described by Bellerose, M et al. [26]. For each experiment, the culture media was prepared daily by thawing the appropriate volume and by diluting it 1:1 with PBS-thioglycolate 0.1%. It was then homogenized for 30 s in a sterile 6 × 9-inch filtra-bag 330 µm (LABPLAS, Sainte-Julie, QC, Canada). The filtrated liquid was transferred in a bottle at 4 °C. As the bigger undigested feed particles were removed by the filtration step to prevent tube clogging, inulin (Fisher Scientific, Ottawa, ON, Canada) was added to the culture media at a concentration of 0.5 g/L to provide the microbiota with non-digestible fiber.

#### 2.5.2. Bioreactor Experiments

Prior to the start of the experiment, all reactors were disinfected with alcohol (70%), with a volume corresponding to 3 times the volume of each reactor. Immediately after, reactors were rinsed with 3 volumes of sterile water. For inoculating the mother reactor, a 20 g aliquot of colon content, prepared as described above, was unfrozen and added 1:1 to freshly prepared culture media. The inoculum was then homogenized in a sterile 10 × 15-inch filter bag of 330 µm (LABPLAS, Sainte-Julie, QC, Canada) for 10 s; 1 mL of the filtrate was flash-frozen in liquid nitrogen for further analysis and the filtrate was added to the mother reactor (R1) under an anaerobic atmosphere. The mother reactor was fed at a flow rate of 22 mL/h of culture media during 24 h in batch mode. After 24 h (T24), 60 mL of the R1 reactor was transferred to each of the daughter reactors (R2 to R6) using a syringe. All reactors were then fed with the culture media at a flow rate of 16 mL per hour in continuous mode. At this moment, to each reactor, 10^6^ CFU of *Salmonella* Typhimurium-R was added using a syringe. The system was then run for 24 h without being further disturbed. At T48, different administration modes of the probiotics were tested. For the reactors R1 and R2, no further modification was carried out during the experiments as they were used as controls.

For the first method of administration (daily), at T48h, reactors 3 and 4 received a volume of PRO-P2702 equivalent to 10^9^ CFU of the probiotic product (as recommended by the manufacturer), using a syringe. This treatment was repeated at T72h and T96h. This condition was used to determine the impact of a daily direct dose of the probiotic on the bioreactor’s resulting microbiota. For the second method of administration (continuous), still at T48h, reactors 5 and 6 received the probiotic that was added directly to the feeding culture media at a concentration of 10^8^ CFU of the probiotic product per mL of culture media, as recommended by the manufacturer. This feeding was maintained for 24 h and then replaced with a regular culture media for the rest of the experiment. This condition was used to simulate the administration of the probiotic to the animal for a continuous 24 h period and to monitor if lasting changes could be observed once the probiotic is stopped. The bioreactor’s assays were made on 4 different weeks (*n* = 4) with two reactors per condition. Each experiment lasted one week.

For each experiment and for each reactor, samples were taken for the study of the bioreactor’s microbiota at the following time point: the inoculum (T0), T24, T48, T72, T96, and T120. Using a syringe, 1 mL from each bioreactor was taken, flash-frozen and stored at −80 °C for sequencing purposes. At the same time, bioreactor samples of 5 mL were also taken to enumerate bacterial populations by culture: total aerobic bacteria, *E. coli*, *Salmonella* Typhimurium-R and *Clostridium* spp. For bacterial enumeration, samples were diluted from 10^−1^ to 10^−7^. A volume of 100 µL of each dilution was then spread on Petri dishes containing the appropriate media: Tryptone Soya Agar with 5% Sheep Blood (Oxoid, Nepean, Canada) for total aerobic bacteria, MacConkey agar (BIOKAR diagnostics, Pantin, France) for *E. coli* (Gram-negative fecal indicator), BGS novo-rifampicin agar for *Salmonella* Typhimurium-R and TSC agar (BIOKAR diagnostics, Pantin, France) mixed with 5% egg yolk emulsion (Neogen, Heywood, UK) and 0.2 mg/mL D-cycloserin (BIOKAR diagnostics, Pantin, France) for *Clostridium* spp. (Gram-positive anaerobic/aerotolerant indicator). After 24 h of incubation at 37 °C, the plates were counted, and the concentration of each bacteria was calculated. All bioreactor samples were kept at 4 °C until DNA extraction.

When no *Salmonella* could be counted, a detection method was enacted on the original bioreactor samples that were kept at 4 °C. Samples were diluted 1/10 in peptone water (BIOKAR diagnostics, Pantin, France), and incubated overnight at 37 °C. Samples were put on an MRSV agar (BIOKAR diagnostics, Pantin, France) that were incubated at 42 °C overnight. On each MSRV, at the migration front, samples were taken and put on BGS novo-rifampicin agar and incubated overnight at 37 °C before being read. Samples were considered positive if typical colonies were observed. Each experiment was repeated 4 times. At the end of each experiment, the reactors were cleaned with 3 volumes of alcohol followed by 3 volumes of quaternary ammonium followed by 3 volumes of sterile water. The reactor was then left to dry at room temperature.

### 2.6. SCFAs Analysis, Total Gross Energy, Total Crude Protein and Dry Matter Analysis

At T120, the end of the experiment, 2 mL of the content of each bioreactor was recovered. Then, 4 mL of H_2_SO_4_ 1.5% (Fisher Scientific, Waltham, MA, USA) were added, then the samples were vortexed. The samples were kept at −20 °C until SCFA analysis was performed. The frozen samples were homogenized and then centrifuged at 10,000 rpm for 15 min at 4 °C. The supernatant was analyzed on a Hewlett Packard 6890N gas chromatograph (Agilent Technologies, Wilmington, DE, USA) equipped with a flame ionization detector and an autosampler (Hewlett, Packard, Avondale, PA, USA) for volatile fatty acid.

The rest of the content of each bioreactor (approximately 40 mL) was individually flash-frozen and kept at −80 °C for digestibility analysis. These samples were analyzed for gross energy (GE, Parr 6300 Calorimeter, Parr Instrument Company, Moline, IL, USA), crude protein (Kjeldahl, method 976.05; nitrogen × 6.25) and dry matter (method 930.15).

### 2.7. DNA Extraction and PCR for DNA Amplification of V4 Region

A volume of 300 µL of flashed-frozen samples was extracted with the PowerLyzer PowerSoil kit (QIAGEN Inc., Toronto, ON, Canada), following manufacturer recommendations and the resulting DNA was stored at −80 °C. The DNA was quantified with a QFX Fluorometer (DeNovix Inc., Wilmington, DE, USA) with the QUBIT BR Assay kit (Thermo Fisher scientific, Waltham, MA, USA). The DNA was then standardized at 10 ng/µL with PCR-grade water. A PCR targeting the V4 section of the 16S rRNA was made in a Mastercycler ^®^Nexus PCR (Eppendorf AG, Hamburg, Germany) using the following primer: 515FP1-CS1F for the forward primer and 806RP1-CS2R for the reverse primer (Life Technologies, Pleasanton, CA, USA) (Appendix A). For each sample, 12.5 ng of DNA was amplified in a final volume of 30 µL using the Invitrogen Platinum Superfi DNA Polymerase kit (Fisher Scientific, Waltham, MA, USA) and Bovine serum albumin 0.4 mg/mL (SIGMA-ALDRICH Co., St-Louis, MO, USA). (Appendix A). The PCR reaction started with 5 min at 95 °C, followed by 23 cycles of 30 s at 95 °C, 30 s at 55 °C and 180 s at 72 °C. When carried out with the 23 cycles, the PCR ended with a 10 min final extension step at 72 °C. PCR products were kept at 4 °C. The positive control was the ZymoBIOMICS Microbial Community DNA Standard (Zymo Research, Irvine, CA, USA) and the negative controls were water submitted to DNA extraction as described above as well as a no template PCR control. To verify the quality of the amplicons, an agarose gel containing SYBR Safe DNA gel stain (Invitrogen, Burlington, ON, Canada) was made and observed under UVs. The samples were sent for Illumina Miseq sequencing at Genome Québec.

### 2.8. Sequence Analysis, Microbiota Diversity and Composition Determination

Raw sequencing reads were demultiplexed, quality-filtered and analyzed using Mothur software version 1.48.0 as previously published with some modifications [26]. Raw reads from the previous study [26] and results from this experiment were combined for the analysis. Rapidly, for make.contigs, pdiffs was set to 2, deltaq to 5, maxlenght to 300 and maxhomop to 20. For alignment, Silva v132 was used. The precluster command was set to a difference of 4 nucleotides. For chimera detection, vsearch was used. Taxonomic assignation was carried out using Silva v138 with the cut-off set at 80 before removing Chloroplast, Mitochondria, unknown and Eukaryota taxons. OTU were formed using opticlust at 3% dissimilarity as cut-off using the cluster.split command set at taxlevel 6. The mothur outputs were then analyzed in RStudio 1.3.1073 using R version 4.0.3. After the importation of the shared and taxonomy files in R, the data were inspected. Controls were removed and the remaining samples were rarefied to the lowest number of sequences within a sample prior to alpha and beta diversity analysis. Shannon and Inverted Simpson indexes were used for the alpha diversity analysis. A Kruskal–Wallis statistical test was used to assess significant differences between conditions. Bray–Curtis dissimilarity index results were visualized with a non-metric multidimensional scaling (NMDS) graph, followed by ADONIS (Vegan package) and Pair-Wise ADONIS (pairwiseAdonis) to test differences between groups. Non-rarefied data were used for Multivariate Analysis by Linear Models (MaAsLin2) using no normalization, no transformation and the analysis method set to NEGBIN to identify specific biomarkers associated with tested conditions.

### 2.9. Statistical Analysis

The ANOVA and *t*-student tests analysis were conducted using Microsoft^®^ Excel^®^ (Version 2306 Build 16.0.16529.20164). When a *p*-value ≤ 0.05 was obtained, the results were considered statistically significant.

## 3. Results

### 3.1. Disk Diffusion Assay of PRO-P2702 against Salmonella Typhimurium, E. coli, Staphylococcus aureus and Enterococcus faecalis

The supernatant of the PRO-P2702 probiotic (Table 1), inhibited the growth of all bacteria starting from the 1/5 dilution and was able to inhibit *S. aureus* and *E. coli* at the 1/10 dilution. As for the pellet (Table 2), it did not inhibit *Salmonella* Typhimurium but inhibited the other three bacteria from the dilution of 1/2. The 1/5 dilution was effective only against *S. aureus* and *E. coli* while the 1/10 dilution inhibited only *E. coli*.

### 3.2. PRO-P2702 Broth Inhibition Test against Salmonella Typhimurium-R Obtained by Cultural Approach in Liquid

PRO-P2702 or the supernatant of PRO-P2702 was tested against two different concentrations of *Salmonella* Typhimurium-R (10^3^ and 10^6^ CFU total) in broth for 120 h, where samples were taken at T10 min, T6h, T24h, and T120h. At the beginning of the experiment, Salmonella Typhimurium-R was present in all the samples to which it was added. After only 10 min of contact time, the totality of *Salmonella* Typhimurium-R in the 10^3^ CFU sample was killed while it took 6 h for the samples inoculated with 10^6^ CFU. The supernatant killed *Salmonella* Typhimurium-R the same way as the whole product (Table 3).

### 3.3. Bacteria Counts in the Bioreactor

According to the t-student statistical analysis (*p* ≤ 0.05), there was no significant difference between the control and the two probiotics administration modes (daily vs. continuous) for *E. coli* and *Clostridium* spp. for any sampled time (Figure 1a,b). At T72, total aerobic bacteria were significantly higher than the control for both modes (Figure 1c). At T96, *Salmonella* was significantly lower than the control for the daily mode and at T120, *Salmonella* was significantly lower than the control for the continuous mode (Figure 1d).

### 3.4. Microbiota Diversity

#### 3.4.1. Probiotic Effect on Microbiota Diversity

At T48, just prior to the start of the experiments, no probiotics were present in the bioreactors; consequently, no statistical differences were found between the control and the two probiotic administration modes for either the alpha diversity (Table 4) or the beta diversity (Figure 2a).

At T72, (24 h after the beginning of treatments for both experiments), no significant differences were observed for the alpha diversity (Table 4), but all the conditions were significantly different for the beta diversity (Figure 2b).

At T96, (24 h after the second treatment for the daily mode and 24 h after the end of the treatment for the continuous mode), the alpha diversity for the continuous treatment was significantly different from the control, but not for the daily mode (Table 4). For the beta diversity, both modes were significantly different (Figure 2c).

At T120, (24 h after the end of the daily treatment and 48 h after the end of the continuous treatment), the alpha diversity for both conditions was significantly different from the control for the observed index. The daily treatment was significantly different from the control using the Simpson index. The continuous treatment was significantly different from the control using the Shannon index (Table 4). For the beta diversity, all the conditions were significantly different (Figure 2d).

#### 3.4.2. Probiotic Effect on the Microbiota Composition

To analyze the microbiota, 6,664,245 clean sequences were obtained. There was a mean of 59,502 sequences per sample. The minimum was 44,889 sequences, and the maximum was 70,817. The main genera found within the microbiota are presented in Appendix A. All results obtained from using MaAsLn2 are also presented in Appendix A.

At T48, there was no genus of family associated with the experiment’s conditions (Appendix A).

At T72, *Veillonellacea_unclassified*, *Lactobacillales_unclassified* and *Escherichia/Shigella* were significantly higher in the continuous condition in comparison to the control. *Enterococcus* was significantly higher, and *Prevotella* was significantly lower than the control for both probiotic conditions. The family of bacteria affected by the addition of the probiotic were *Veillonellaceae* and *Enteroccocaceae*, which were significantly higher, and *Anaerovoracaceae*, which was significantly lower in the continuous condition in comparison with the control. For the daily dose, the families *Ruminococcaceae* and *Bacteroidaceae* were significantly lower, and the family *Enteroccocaceae* was significantly higher in comparison with the control (Appendix A).

At T96, many genera were significantly lower or absent in the continuous treatment in comparison with the control, namely the *Prevotellaceae*, *Olsenella*, *Lachnospiracea*_*unclassified*, *Oribacterium*, *Fusicatenibacter*, *Anaerovoracaceae Mogibacterium*, *Collinsella*, *Blautia* and *Prevotella*. For the daily dose, only the genus *Ruminiclostridium* was significantly lower than the control. The loss of many genera for the continuous condition means the loss of many corresponding families such as *Lachnospiraceae*, *Bifidobacteriacea*, *Erysipelotrichaceae*, *Eggertjellaceae*, *Coriobacteriaceae*, *Muribaculaceae* and *Prevotellaceae* that were all significantly lower than the control. The family *Enterococcaceae* was the only one with a significantly higher difference for the continuous condition in comparison with the control (Appendix A).

At T120, the genera *Butyricoccus*, *Agathobacter*, *Muribaculaceae*, *Coprococcus* and *Blautia* were all significantly lower than in the control for the continuous condition. Only the genus *Dorea* was significantly higher than the control. For the daily dose condition, the genus *Coprococcus* was significantly lower than the control. The genera *Anaerovibrio* and *Dorea* were significantly higher than the control. The families *Eggerthellaceae*, *Prevotellaceae*, *Atopobiaceae*, *Peptococcaceae*, *Muribaculaceae* and *Ruminococcaceae* were all significantly lower than the control for the continuous condition. The families *Fusobacteriaceae* and *Enterococcaceae* were significantly higher in the continuous condition compared to the control. There were no significant differences in the daily dose condition for the family (Appendix A).

#### 3.4.3. Investigation of the Alteration in Microbial Composition in the Bioreactor under Different Conditions

The evolution of the control bioreactor microbiota was compared (Study 2) to what was obtained in a previous study (Study 1) [26]. Both studies used the same system, but the following modifications to the original design were made in an attempt to improve the model for Study 2: the mother reactor was filled for 24 h instead of 6 h, a mix of anaerobic gas was used instead of pure nitrogen during the first 24 h, the culture media were bubbled with the gas 30 min prior to first use, inulin was added to the culture media, and the culture media were pooled and distributed into fraction when made and 20 g of inoculum was used instead of 10 g. Compared with the first study made with this system, the microbial diversity was closer to the piglet feces inoculum (Figure 3).

For this study, the evolution in time of the reactor microbiota was then assessed. Time influenced the beta diversity of the piglet’s microbiota (Figure 4). Indeed, the microbiota was distinct at T0 and T24, then it evolved into a much more stable microbiota starting at T72, as T72, T96 and T120 were not different. This observation also applied when only the control samples were used (Figure 5).

The alpha diversity from the control reactor shows no significant differences between the piglets and the T0 condition and between conditions T24, T48, T72, T96 and T120 (Figure 6).

### 3.5. Total Gross Energy and SCFAs

Energy, crude protein, and dry matter concentration were not different for both conditions in comparison with the control (Appendix A). For the SCFA, there was no significant difference between the daily dose treatment and the control for all the SCFAs tested. The continuous dose showed significantly lower acetate and total volatile fatty acid (VFA) in comparison with the control (Figure 7a,b, respectively). When using proportions (%), a significant decrease in acetate and an increase in propionate was observed for the continuous condition as well (Figure 7c,d).

## 4. Discussion

The present study provided new insights into the effect of a probiotic-based product in reducing *Salmonella* and modulating pig-derived intestinal microbiota using a bioreactor.

Using the disk diffusion assay, the results of the present study showed that the probiotic bacteria within PRO-P2702 (in the pellets, which is composed of a mixture of *Bacillus* spp. strains), as well as the probiotic bacteria’s products (in the supernatant, comprising exo-enzymes and bacteriocins from the bacteria and unknown product for stabilizing the solution), were associated with inhibitory effect against a large spectrum of bacteria present in the intestinal microbiota of piglets apart from *Salmonella* where the inhibitory effect of the probiotic seems to come from the probiotic’s by-products and not by the *Bacillus* themselves, as observed in other studies [21,35,36]. This might be due to a variability in the inhibitory effect of *Bacillus* spp. on pathogens [37].

The broth assay was then performed to verify that the inhibition of *Salmonella* Typhimurium was bactericidal or bacteriostatic as well as to assess the inhibition on different inoculating concentrations of *Salmonella*. In fact, bactericidal activity was observed for the tested *Bacillus*-based product in 6 h or less, depending on the *Salmonella* inoculum as observed in other studies [35,36].

Even if the growth competition with the *Bacillus* strains did not show inhibition of *Salmonella* for the disk diffusion assay in the present study, the supernatant demonstrated inhibitory effects against *Salmonella*. It was therefore decided to pursue the bioreactor model. Bacteriology was first used to count indicator bacteria that allowed a simple and relatively fast monitoring (compared to sequencing) of the bioreactor’s microbiota. *Clostridium* spp. exhibited a major drop from T24 to T48 that may have been caused by the presence of oxygen during the transfer at T24. Concerning the total bacteria count, the increase at T72 of total bacteria in the two conditions compared with the control is probably due to the addition of the *Bacillus*-based product but no increase could be observed in the 16S sequencing, which is contradictory. This might be due to problems with the DNA extraction despite the fact that the kit used should have been able to lyse the spores [38]. In the present study, the results between the in vitro and the bioreactor culture-based experiments were different which appeared to be the case in some other studies [17,39]. This was expected as working on the community level is different than working on the strain.

For the alpha diversity, the continuous treatments lowered richness more than the daily dose condition even after the product was removed from the bioreactor (Table 4). The *Bacillus*-based product used in our study seemed to be more potent in the diminution of the alpha diversity compared to other studies, especially for the continuous mode of administration [40,41]. This finding could be explained by the duration of the experiment, which was short, the compounds present in the *Bacillus*-based product, which could bear a strong antimicrobial activity, or the possible synergy between the probiotic secreted antimicrobial compounds or enzymes that increased the microbiota modulation. The loss of alpha diversity, especially for the richness, could induce dysbiosis in an in vivo model [42]. It therefore seemed that a lower dose should be used for the continuous mode of administration.

For the beta diversity results, both modes of product administration modified the microbiota structure, and the continuous treatment seemed to have a different effect on the microbiota compared to the daily dose treatment. After 24 h of treatment, the continuous treatment samples differentiated more from the control when compared to the daily dose treatment, and this remained true over time (Figure 2). Some studies showed no significant change in beta diversity in the gut microbiota when administrating a *Bacillus*-based probiotic to swine [41,43], but others did [40,44]. This might be due to the length that the treatments were administered, the mode of administration, or a difference in the product tested.

MaAslin2 analysis was then used to reveal biomarkers associated with the different treatments. Since time T48 involved no different conditions, there should be no differences between reactors. Statistically, alpha and beta diversity were the same (Figure 2; Table 4). However, few differences at T48 were observed (Appendix A). First, an unclassified genus in the *Prevotella* family was significantly increased in the control. A bacteria from the *Succinivibrionaceae* family was also significantly lower for the continuous condition at T48. This family of bacteria is usually recognized as a dietary fiber digestor [45].

At T72, 24 h after the start of both modes of product delivery, in the continuous condition, an increase in the family *Veillonellaceae* and the order of Lactobacillales was observed. *Veillonella* spp. are bacteria that are normally present in the intestinal tract and are associated with healthy gut microbiota because of their ability to produce SCFAs [46]. *Veillonella* also showed an effective inhibitory effect against *Salmonella* spp. in vitro and in vivo [8,47,48,49,50,51]. As for the augmentation of the Lactobacillales order, some genera being part of this order such as *Streptococcus* and *Lactobacillus* are decreased when in the presence of *Salmonella* [8,49]. The increase in the Lactobacillales order correlated with the increase in the family *Enterococcaceae*. In the daily dose treatment, the genus *Enteroccocus* and the family *Enterococcaceae* were also increased. These bacteria are commensals of the microbiota and can be used as probiotics [52]. The family *Ruminococcaceae* and the genus *Prevotella* were decreased. These bacteria usually relate to a healthy intestinal microbiota even if in some rare cases, they could cause opportunistic diseases in humans [53,54,55]. Moreover, *Ruminococcaceae* can metabolize complex polysaccharides into useful nutrients for the host [54]. These bacteria are negatively correlated with the presence of *Salmonella* [48]. The decrease in theses populations in the present study could be attributed to the effect of the *Bacillus*-based product which could have an inhibitory effect against them.

Starting at T96 and up until T120, there was a loss of many genera and families of bacteria with the continuous treatment, an observation supported by the lower alpha diversity in this condition. This loss of diversity was less drastic with the daily dose treatment. In fact, for the continuous treatment, a loss of seven families and nine genera was observed at T96 (compared to T72) while a loss of six families and five genera was observed at T120 (compared to T72). For the daily treatment, only the loss of one genus was observed for both times. Furthermore, many of the genera lost in the continuous treatment were from the *Lachnospiraceae* family. This family has been associated with a *Salmonella*-free microbiota [48,51]. The genus *Blautia* was also decreased for continuous treatment at T96 and T120 and is mostly considered for its potential to produce SCFA and promote gut health [56].

The bioreactor in the present study was initially used to monitor the impact of essential oils in a piglet’s derived intestinal microbiota [26]. Modifications to the system resulted, in the first 48 h, in environmental conditions aiming to generate conditions closer to the piglet’s intestinal microbiota than in the Bellerose et al. study [26]. These results indicate that the changes made in this study were effective in improving the microbiota survival conditions in the system.

Concerning the SCFAs analysis, the results obtained in this study showed, for the continuous dosing, a decrease in acetate and total volatile fatty acid concentrations, a decrease in the proportion of produced acetate but an increase in propionate proportion. A decrease in SCFA production when using probiotics has also been observed in another study [45]. This is contradictory to the fact that probiotics in general usually increase the production of SCFAs such as propionate, acetate, and butyrate [57]. *Bacillus* spp. Mostly produce lactic acid, acetate, propionate and butyrate but can also produce iso-butyrate, valerate and iso-valerate, depending on the species and the strain [58]. They were also shown to modulate the microbiota by increasing SCFA producer relative abundance in intestinal microbiota such as *Lactobacillaceae*, *Ruminococcaceae*, and *Lachnospiraceae* [58,59]. The decrease in SCFA in the continuous condition could be related to the fact that some of these families were lost at T96 and T120 as seen in the MaAslin 2 analysis. Propionate, acetate and butyrate are recognized as *Salmonella* inhibitors [60]. Their presence, especially the increase in the propionate proportion, may contribute to the *Salmonella* inhibition observed.

Probiotics are well known to increase digestibility by helping to metabolize proteins [61]. However, our study showed no difference in crude protein proportions for both treatments. It agrees with some other studies that showed no difference in crude protein concentration [62,63]. But is in contradiction with other publications [64]. Energy and dry matter are also parameters that may vary depending on the study. Here, again, results differ between already-published studies [65,66]. The results from our study suggest that little to no increase in digestibility was achieved from the probiotic action. It is important to note that in animals, absorption occurs in the intestinal tract while no absorption is simulated in the bioreactor system used in the present study. It is also important to note that digestibility analysis was performed, due to technical constraints, at the very end of the experiments, when no probiotics were added into the reactors. Therefore, it seems that the probiotic should be present at all times for it to potentially impact digestibility.

With the use of the bioreactor, we were able to show that the probiotic-based product had an impact on *Salmonella* and the microbiota derived from piglets. There are some limitations associated with this system. Indeed, this system does not consider the interaction of the microbiota with the host, especially for the immune components. Probiotics can also improve the host’s immunity which could further inhibit *Salmonella* [67]. Moreover, not all bacterial species are retained in bioreactors compared to the inoculum, and some of them might be important for animal health or pathogen inhibition. Nevertheless, since an effect on *Salmonella* and on the microbiota was observed in the bioreactor, and the daily dose did not cause major disruption on the microbiota, it greatly increased the confidence that the product would work in live animals and that it is worth trying in vivo.

## 5. Conclusions

This study showed that the PRO-P2702 *Bacillus*-based product has an inhibitory effect on *Salmonella* and a modulation effect on piglet-derived colon microbiota in a bioreactor system. It was also shown that the mode of administration of the *Bacillus*-based product had different impacts on the microbiota. Since the continuous dose reduced greatly the alpha diversity, reduced the abundance of some bacterial populations considered healthy for the animals, and reduced total SCFA production, this way of administration or the dose employed may not have been optimal and could be revisited. Nevertheless, since an effect on *Salmonella* and on the microbiota was observed in the bioreactor for the daily dose, it would be interesting to validate those observations in an in vivo study. Giving pig producers access to another solution to increase food safety and the health of their herd is crucial in increasing the sustainability of the sector.

## Figures and Tables

**Figure 1 animals-14-00787-f001:**
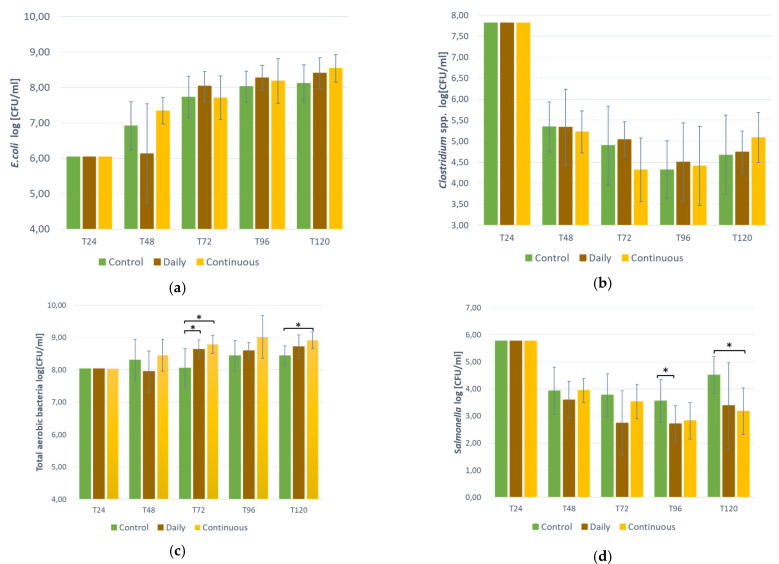
Bacteria count in the bioreactor in presence of the probiotic PRO-P2702 at different time points: (**a**) *E. coli*; (**b**) *Clostridium* spp. (**c**) Total aerobic bacteria (**d**) *Salmonella* Typhimurium-R. Significant when *p* < 0.05 using student *t*-test after ANOVA test. *: statisticaly significant.

**Figure 2 animals-14-00787-f002:**
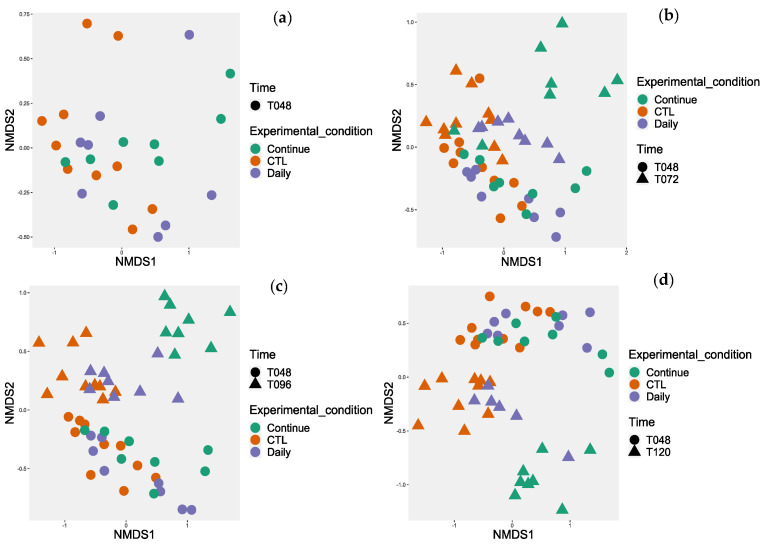
Non-metric multidimensional scaling (NMDS) representation of the beta diversity on the bioreactor’s microbiota according to probiotic’s modes of administration and time. The Bray–Curtis index was used. The pairwise ADONIS statistical test showed a significant difference between all the groups (Continue vs. CTL vs. Daily) (*p* < 0.05) except at T48. Where no differences were observed. (**a**) At T48 (**b**) Comparison between T48 and T72 (**c**) Comparison between T48 and T96 (**d**) Comparison between T48 and T120.

**Figure 3 animals-14-00787-f003:**
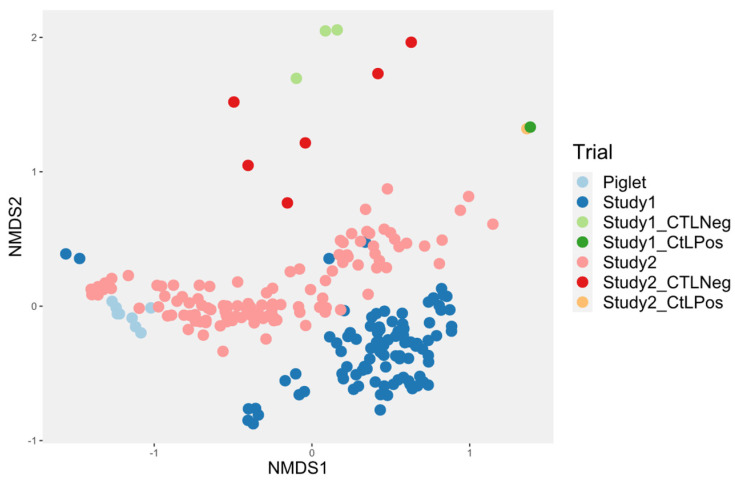
Comparison of the beta diversity according to time between the two studies using the same bioreactor system. CTLNeg = water, CtlPos = artificial community.

**Figure 4 animals-14-00787-f004:**
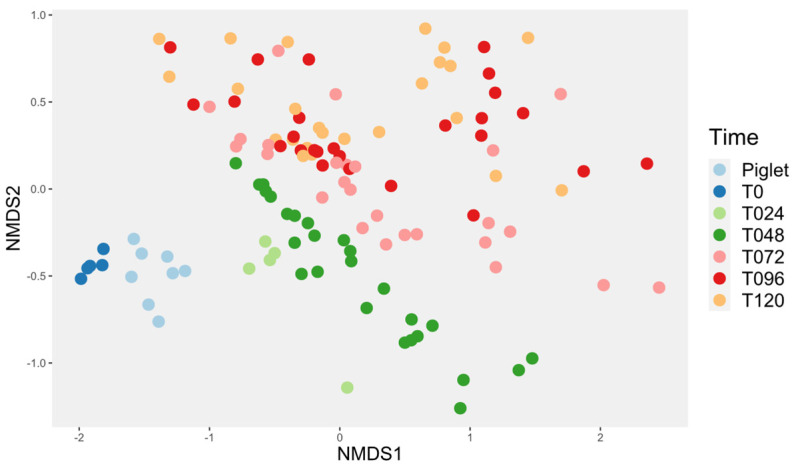
Time evolution of the bioreactor’s microbiota compared to piglet’s feces. The Bray–Curtis index was used. The pairwise ADONIS statistical test showed a significant difference between all the groups (*p* < 0.05) except for T072 vs. T120 (*p* = 0.07), T096 vs. T120 (*p* = 0.83) and T072 vs. T096 (*p* = 0.41).

**Figure 5 animals-14-00787-f005:**
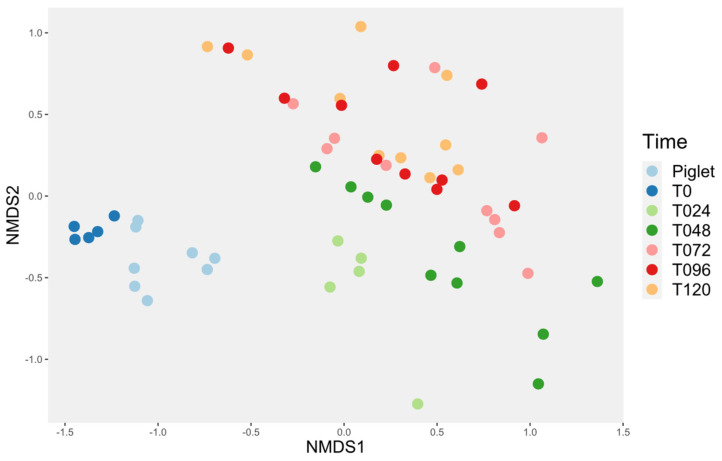
Evolution of the bioreactor control microbiota. The Bray–Curtis index was used. The pairwise ADONIS statistical test showed a significant difference between all the groups (*p* < 0.05) except for T072 vs. T120 (*p* = 0.07), T096 vs. T120 (*p* = 0.82) and T072 vs. T096 (*p* = 0.41).

**Figure 6 animals-14-00787-f006:**
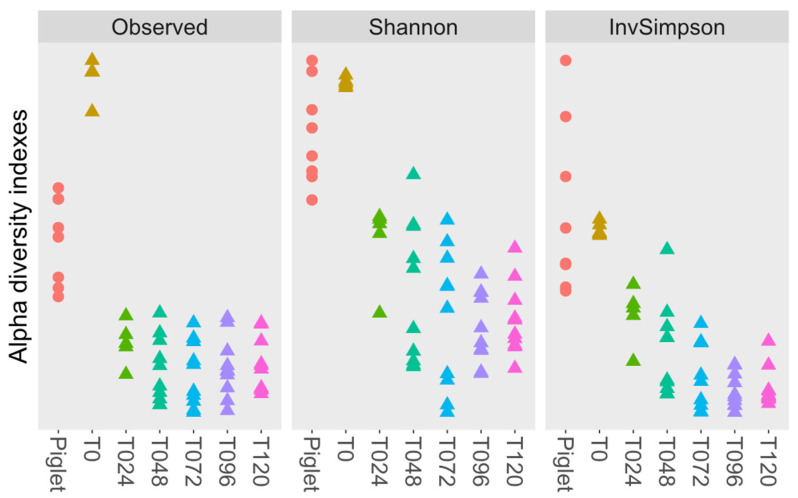
Alpha diversity evolution for the control microbiota. The Kruskal–Wallis statistical test followed by a pairwise student-*t* test showed no significant difference between Piglet and T0 and between T24, T48, T72, T96 and T120 (*p* > 0.05) due to microbiota stabilization.

**Figure 7 animals-14-00787-f007:**
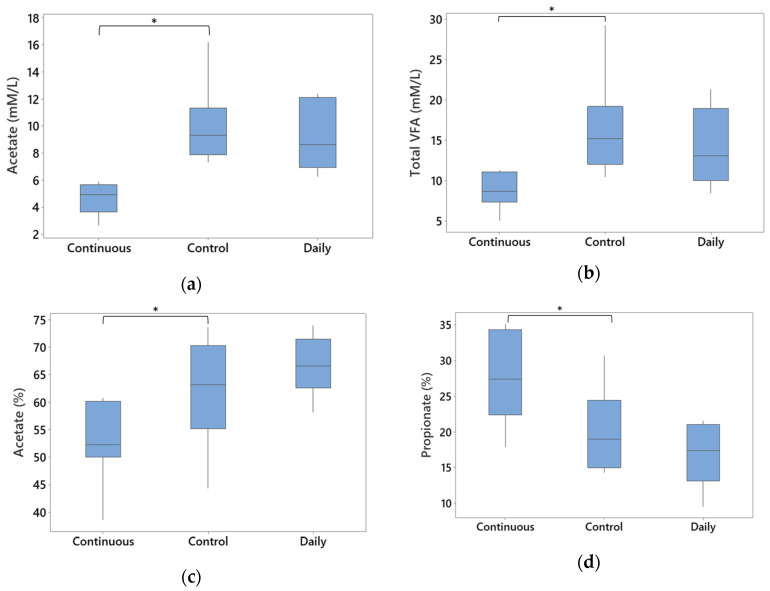
Significant changes in short-chain fatty acid production in millimolar/milliliter (mM/mL) or in proportion (%). ANOVA test followed by student *t*-test showed significant results. (*p* ≤ 0.05 (*)) (**a**) Concentration of acetate in all conditions at T120 (**b**) Concentration of total volatile fatty acids in all conditions at T120 (**c**) Proportion of acetate in all conditions at T120 (**d**) Proportion of propionate in all conditions at T120.

**Table 1 animals-14-00787-t001:** Inhibition diameter with the supernatant of PRO-P2702 (*n* = 1).

		Dilution of the Supernatant				
Inhibition diameter (mm)	Strains	Not diluted	1/2	1/5	1/10	1/100
	*Salmonella* Typhimurium	11	10	9	0	0
	*Staphylococcus* *aureus* ATCC 25923	25	21	15	10	0
	*Enterococcus faecalis ATCC 29212*	21	17	10	0	0
	*Escherichia coli ATCC 25922*	13	11	8	7	0

**Table 2 animals-14-00787-t002:** Inhibition diameter with the pellet of PRO-P2702 (*n* = 1).

		Dilution of the Pellet				
Inhibition diameter (mm)	Strains	Not diluted	1/2	1/5	1/10	1/100
	*Salmonella* Typhimurium	0	0	0	0	0
	*Staphylococcus aureus* ATCC 25923	26	20	13	0	0
	*Enterococcus faecalis ATCC 29212*	21	17	0	0	0
	*Escherichia coli* *ATCC 25922*	17	13	11	11	0

**Table 3 animals-14-00787-t003:** Presence or absence of *Salmonella* Typhimurium-R with 10^10^ CFU of the PRO-P2702 whole product or the supernatant only, according to contact time.

Conditions (in CFU)	T0	T10 min	T6, 24 and 120 h
PRO-P2702 whole + 10^6^ *Salmonella* Typhimurium	Present	Present	Absent
PRO-P2702 whole + 10^3^ *Salmonella* Typhimurium	Present	Absent	Absent
PRO-P2702 supernatant + 10^6^ *Salmonella* Typhimurium	Present	Present	Absent
PRO-P2702 supernatant + 10^3^ *Salmonella* Typhimurium	Present	Absent	Absent
10^6^ *Salmonella* Typhimurium control	Present	Present	Present
10^3^ *Salmonella* Typhimurium control	Present	Present	Present
PRO-P2702 control	Absent	Absent	Absent
Negative control	Absent	Absent	Absent

**Table 4 animals-14-00787-t004:** Alpha diversity analysis.

	Control			Daily			Continuous		
Time	Obsv	Sh	Isimp	Obsv	Sh	Isimp	Obsv	Sh	Isimp
T048	232	2.8	10.3	200	2.5	10.3	198	2.4	7.4
T072	240	2.6	7.7	170	2.3	6.6	172	2.0	4.8
T096	242 ^a^	2.6 ^d^	6.4 ^f^	184 ^a^	2.4 ^d^	5.7 ^f^	138 ^b^	2.0 ^e^	5.0 ^g^
T120	238 ^a^	2.6 ^d^	6.7 ^f^	175 ^b^	2.4	5.6 ^g^	124 ^c^	2.1 ^e^	4.6

Obsv = Observed index, Sh = Shannon index, Isimp = Inverse Simpson index (^a–g^ = significantly different (*p* ≤ 0.05)).

## Data Availability

Data are contained within the article and Appendix A and can be provided upon reasonable request.

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
