# Peer review of "Characterization of the Effects of a Novel Probiotic on Salmonella Colonization of a Piglet-Derived Intestinal Microbiota Using Improved Bioreactor"

_animals, 2024, doi:10.3390/ani14050787_

Round 1

Reviewer 1 Report

Comments and Suggestions for Authors

1. For the probiotic bacteria (PRO-P2702) the authors used in the study, please provide the viable number of Bacillus in the product and source of isolation if information is available;

2. For centrifugation, please convert "rpm" to "x g";

3. Line 142, what do you mean by "saline water", use "saline" instead;

4. Figure 1, please mark significance between values in the chart using letters or *;

5. Please revise table title to indicate experimental design, other information (e.g., explanation of abbreviations) should be provided as table footnote;

6. Figure 2, please increase the resolution of this figure and include the ADONIS analysis results in each panel in the Figure and Figure 3. marks for each panel (e.g., A, B, C, D) should be on the left. Please explain the information of each panel in the figure legends;

7. Please increase the resolution and font size of supplemental Figures;

8. Please revise the figure legend for Supplemental Figures 2-9 to explain the detail of the analysis;

9. In figures, please use only one expression for "daily" and "unique", please revise;

10. For the comparison of the relative abundance of the bacteria at General and Family level, did you normalized the sequence data? Please explain in the Materials and Methods section.

Comments on the Quality of English Language

Minor editing of English language is required to further improve the quality of the manuscript.

Author Response

Thank you for your comments !

Reviewer 2 Report

Comments and Suggestions for Authors

Grandmont et al. present in their manuscript a methodological improvement of a bioreactor to study piglet intestinal microbiota, and use it to evaluate the impact of a Bacillus-based probiotics blend as anti-Salmonella treatment. To this end, piglet microbiota is grown in the bioreactor, followed by addition of Salmonella and subsequent treatment with or without the Bacillus-based probiotics blend. Additional culture-based assays are used to evaluate the growth inhibition of the blend and its supernatant on various bacterial strains.

The data presented are of good technical quality and the interpretation and conclusions are globally sound.

Some key elements are however missing in terms of information, and/or interpretation:

Globally the text is too long. Parts of the method section could be moved to the results as they would be better placed there for ease of reading (e.g. line 388 where it would be advisable to summarize that the results are obtained by cultural approach in liquid).

Line 129: The composition of the blend should be mentioned, with indication in CFU and relative proportion of the various probiotics. Note that according to my search, this probiotic is not present on NUVAC Eco-Science intranet page, what does it mean (not yet commercially available)?

Line 342: details on sequencing (kits used, read length, number of reads/sample etc..should be provided here (even if some is the results section such as reads/sample).

Figure 2 is of low quality and should be improved. Legend to explain panels A to D should be added.

Paragraph 3.4.2: there is no mention of any attempt to detect the various Bacillus probiotics when added. Such information would be needed and highly relevant.

Lines 479 to 481 (details of sequencing data) should be moved at the beginning of paragraph 3.4.2 for logical reading.

Sentence from line 481 (“The evolution..) to line 489 is way too long. Consider rephrasing for sake of clarity.

Globally paragraph 3.4.3 should include a comparison with / without probiotics addition to evaluate global effect on piglet microbiome and complement figure 2 description.

Discussion is globally too long. Consider rephrasing and shortening.

Second paragraph: Line 546: don’t we expect and effect even mild o the pellet?

Line 567: Globally, one approach that could be envisaged to understand the various impact of different Bacillus would be to perform comparative genomics as most of the genomes must be available. Such proposal would benefit the discussion.

Line 600: the kit used is specified to extract Bacillus DNA, please comment, and adapt the text.

Lines 601-602: unclear sentence, consider rephrasing.

Line 608-620: what can be concluded from this paragraph?

Line 694: if the family described is associated with Salmonella free microbiome, what can it mean mechanistically regarding the study presented here? What about comparing control samples (no Bacillus-blend) compared to samples with the blend?

Information described in paragraph lines 702-720 could be reflected in the title (e.g. …using improved bioreactor…).

What is the purpose of figure supp1?

Figure supp 16: what does unique refer to. Use same terms (daily?) as in the rets of the paper for clarity.

Minor text edits:

Line 35: Illumina sequencing is not per se a microbiome analysis. Reformulate with inclusion of the concept of microbiome profiling by targeted 16S rRNA gene sequencing (on Illumina).

Line 114: rephrase for more logical sense: microbiota diversity within the bioreactor using 16S rRNA amplicon sequencing (31)

Line 521: punctuation missing before “When..”

Line 899: Reference form the authors (!) must be cited with journal name issue etc…

The study in its current form is not ready to justify publication.

Comments on the Quality of English Language

English adequate. Some sentences too long or unclear and must be rephrased.

Author Response

Thank you for your comments !

Reviewer 3 Report

Comments and Suggestions for Authors

This study by Grandmont et al investigates the effects of a Bacillus probiotic formulation on enteric pathogens growth by using in vitro experiments and a bioreactor. While some findings are valid, following comments have to be addressed to make this study suitable for publication. 

1. In Simple summary, please provide a statement of the main finding/focus of this study. 

2. In line 122, please include PRO-P2702 along with the wording BAcillus-based liquid product, since you mention PRO-P2702 a lot later, and is not introduced separately elsewhere in the results section.

3. line 130-131, I did not understand what is pig production? Like breeding pigs? 

4. line 141: grammatical correction, it is prior "to" being used.

5. Table 2 and Table 3: How many times were the in vitro experiments repeated, there is no details about n numbers. And, what statistical tests were performed to check if the antimicrobial activity is significant or not. Is it significant? No information about this is provided. 

6. line 388-392: Even though details on broth inhibition test is described under methods section, please proved a line on what was tested here, since the results in this section appear confusing without this information. 

7. Figure 1 would be easier to follow if Y axis is labelled with bacteria or treatment condition included. For et, E. coli log[CFU/ml] etc. 

8. In Table 5, no stats information is provided for 2.4 and 4.6 at T120. Is it d/e or g/h.

9. lines 425-428, At T72, CTL and daily seems clustered together. Where is the beta-diversity significance test results? Write the p values on this figure or provide a supplementary table. 

10. Supplementary figures 4,5,6 and 7 are pixelated, low resolution and difficult to read the axes and labels, please provide better figures. 

11. In supplementary figure 2 and 3, if nothing is significantly different why are specific genus or family exclusively compared in those graphs, not clear and no explanation given. 

12. The title "evolution of the microbiota" for 3.4.3 is vague, since this study does not exactly track evolution, it investigates alteration in microbial composition in the bioreactor under different conditions. So please use a more appropriate title. 

13. In Figure 3, what is CTLNeg and CTLpos. It is not clear and no explanation is provided. 

14. I don't understand how Figure 4 and Figure 5 are different from the text description or by looking at the figures, please explain.

15. In Figure 6, isn't there a difference between To and T24, why is not compared or discussed. 

16. Figure 7 and Figure 8 are essentially discussing the same things. I would suggest combining them. 

17. Discussion sections seems very exhaustive and a lot of it unnecessary, because of which relevant information is lost or lacking. I would advise rewriting it, please mention the main findings- and primarily try to explain what would have caused a difference in the in vitro studies versus bioreactor experiments, and within bioreactor experiments why is there a difference in continuous versus daily dose, and lastly why is the observation from bioreactor experiment important and it's possible application.

Comments on the Quality of English Language

English seems overall fine.

Author Response

Thank you for your comments !
